# Bio-Inspired 3D Affordance Understanding from Single Image with Neural Radiance Field for Enhanced Embodied Intelligence

**DOI:** 10.3390/biomimetics10060410

**Published:** 2025-06-19

**Authors:** Zirui Guo, Xieyuanli Chen, Zhiqiang Zheng, Huimin Lu, Ruibin Guo

**Affiliations:** College of Intelligence Science and Technology, National University of Defense Technology, Changsha 410073, China; guozirui@nudt.edu.cn (Z.G.); xieyuanli.chen@nudt.edu.cn (X.C.); lhmnew@nudt.edu.cn (H.L.)

**Keywords:** neural radiance fields, 3D affordance models, robotic manipulation

## Abstract

Affordance understanding means identifying possible operable parts of objects, which is crucial in achieving accurate robotic manipulation. Although homogeneous objects for grasping have various shapes, they always share a similar affordance distribution. Based on this fact, we propose AFF-NeRF to address the problem of affordance generation for homogeneous objects inspired by human cognitive processes. Our method employs deep residual networks to extract the shape and appearance features of various objects, enabling it to adapt to various homogeneous objects. These features are then integrated into our extended neural radiance fields, named AFF-NeRF, to generate 3D affordance models for unseen objects using a single image. Our experimental results demonstrate that our approach outperforms baseline methods in the affordance generation of unseen views on novel objects without additional training. Additionally, more stable grasps can be obtained by employing 3D affordance models generated by our method in the grasp generation algorithm.

## 1. Introduction

Affordance was first proposed by Gibson [1], referring to the ability of the environment to contain information relevant to interactions with the organisms. Since then, the concept of affordance has been widely used in the field of robotics [2,3]. Embodied intelligence is an important research topic in robotics, with extensive studies focusing on achieving stable and accurate grasping and manipulation. For specific objects like screwdrivers, drills, and other tools, considering their structure is crucial for performing appropriate manipulation beyond simple grasping. The different structures of the object are represented by various semantics in the image. Such specific semantics relevant to manipulation can be treated as affordance.

Although machine learning methods [4] have proven valuable in semantics prediction, which is relevant to affordance generation, they may suffer when handling the different structures of the object. Moreover, the algorithm cannot provide a complete 3D model of the object along with the prediction. Generating appropriate 3D affordance understanding for robotic manipulation is still challenging.

To tackle this, neural radiance fields (NeRF) [5] have been introduced as implicit representations that have achieved success for 3D scenario representations of geometry and texture. These methods are trained with posed images and can generate a 3D model of objects. Existing research focuses on enhancing rendering speed and enabling controllable synthesis to improve implicit representation. However, the exploration of unified affordance interpretation of homogeneous objects in neural rendering remains limited.

Unlike semantic understanding at the object class level [6], affordance often resides in specific object structures and can vary depending on object types. This requires the neural implicit representation to generate diverse syntheses controllably in response to various object categories. It is extremely difficult to control the synthesis directly since the neural network has extensive parameters. Bio-inspired by the neural mechanisms underlying human affordance understanding, this paper addresses the problem for specific category objects. Cognitive neuropsychology studies reveal that the left inferior parietal lobule plays a key role in processing seen-category objects, where such recognition critically depends on prior knowledge of shape and appearance [7,8]. Motivated by this functional segregation, we design AFF-NeRF with a dedicated shape and appearance feature extraction module, mimicking the specialized processing of the inferior parietal lobule. This module specifically addresses the 3D affordance understanding of objects within distinct categories.

Our main contributions are threefold:We design AFF-NeRF with a shape and appearance feature encoder to obtain the 3D affordance models. Our method can produce precise affordance with a single view of novel objects without any retraining or fine-tuning.We created a new dataset for affordance rendering validation, comprising different objects with diverse shapes in various poses and affordance annotations.The affordance models obtained by our approach can effectively improve the performance of downstream robotic manipulation tasks. We evaluate the performance of various grasp generation algorithms, and the results show that applying the affordance models can generate more stable grasps.

## 2. Related Work

Controllable synthesis rendering with NeRF: Implicit neural representation-based controllable synthesis generation can be broadly grouped into two aspects. One is to leverage the generative model framework to output synthesis by controlling the latent code [9,10,11,12,13,14], and the other is working in a supervised learning manner using various input images [15,16,17]. GRAF [12] is one of the early works to combine generative models with implicit representations, which has become the baseline for the subsequent work. Based on GRAF, GIRAFFE [9] decouples different objects and the background to arrange various latent codes. Ultimately, this method makes it possible to generate various objects with different poses. To be more user-friendly, semantic-driven generation with NeRF has been proposed [14,18]. Synthesis generation based on the generative model is more applicable to artistic creation, data provision, etc., and is not appropriate for generating specific objects. Liu et al. [16] uses a supervised learning method to generate various homogeneous objects. However, this method can only work on the training set and, therefore, does not have the ability to generalize to new objects. Jang et al. [15] propose an auto-decoder architecture with decoupled shape and appearance of latent spaces. For new objects, the method requires additional retraining.

Our approach differs from existing methods, and the comparison is shown in Table 1. We present a novel controllable affordance generation pipeline. Our method expands the representation to include operable affordance and demonstrates its ability to generalize to unseen objects with a single image without any retraining or fine-tuning.

Affordance-based robot manipulation: To achieve autonomous manipulation, researchers have conducted extensive studies in perception-based methods [19,20,21]. These studies use original point clouds and RGB images as input, focusing on analyzing the geometry of the object, while using original sensor data may struggle with structurally complex objects [22]. To address this issue, Appius et al. [22] segment the structures of objects in the image and input the result of affordance understanding into the grasp generation algorithm to achieve a more stable grasp. With advancements in computer vision technology, obtaining 3D models of objects has become easier, enabling the generation of higher-quality manipulation strategies based on object models [23,24].

## 3. Methodology

We aim to design a controllable affordance generation pipeline that can handle both known and unseen objects. Figure 1 contains an overview of our method. We exploit deep residual networks to extract the shape and appearance codes and feed them into our AFF-NeRF, which is then trained with affordance labels. Meanwhile, a dataset containing various categories of objects is built to support our training and testing (Section 3.2).

### 3.1. Network Architecture and Training

Given a single image *I* from *n* images ($I_{1}∼I_{n},n\geq 1$), we extract the shape and appearance feature using two deep residual networks, ResNet18 [25], for each image. The shape and appearance features of various images are then passed to the linear layer for integration to output two vectors, which we denote as shape code $C_{s}$ and appearance code $C_{a}$:(1)$$C_{s}=SE\left(I\right),C_{a}=AE\left(I\right),$$
where SE and AE are the shape encoder and appearance encoder, as depicted in Figure 2.

Our goal is to get implicit representations of homogeneous objects with affordance. Since most current NeRF methods cannot output affordance, we design AFF-NeRF with affordance output for homogeneous objects. Inspired by [6], we let the network output the affordance in combination with the volume density. On the front end of AFF-NeRF, we add a multi-layer perceptron (MLP) that integrates the shape and appearance codes, as shown in Figure 2.

More specifically, the shape and appearance codes are fed into the AFF-NeRF to adapt to various objects. As introduced in previous work [5], the NeRF network *F* has two stages of output $F_{xyz}$ and $F_{\theta \phi }$: the former is view-independent, which is related to the geometry of the scene, and the latter is view-dependent, which is related to the texture of the scene.(2)$$\begin{array}{c} F=F_{xyz}⊙F_{\theta \phi }. \end{array}$$

The shape code $C_{s}$ serves as the key to control the geometry. Thus, we combine it with (x,y,z) after position encoding to output volume density σ and affordance *a*, which are strongly correlated with the geometry.(3)$$\begin{array}{c} F_{xyz}:\left(PE\left(x,y,z\right),C_{s}\right)\to \left(\sigma ,a,q\right), \end{array}$$
where *q* is the intermediation. The dimension of *a* is determined by the object structure. Since the appearance code $C_{a}$ is responsible for the texture, we combine it with (θ,ϕ) after position encoding to generate color *c*.(4)$$\begin{array}{c} F_{\theta \phi }:\left(PE\left(\theta ,\phi \right),C_{a},q\right)\to \left(c\right), \end{array}$$
The whole procedure can be formulated as:(5)$$\begin{array}{c} F:(PE\left(x,y,z\right),PE\left(\theta ,\phi \right),C_{s},C_{a})\to \left(\sigma ,c,a\right). \end{array}$$

We follow the volume rendering method [6] to render the image with color and affordance pixel-by-pixel. Given the pixel and the focal point of the camera, we can obtain a ray: r(t)=o+td. Sample K random points ${\left\{t_{k}\right\}}_{k=1}^{K}$ on this ray between near and far bounds ($t_{n}$ and $t_{f}$). The estimated color C(r) and affordance A(r) of the pixel can be expressed as:(6)$$\hat{C}\left(r\right)=\sum_{k=1}^{K}\hat{T}\left(t_{k}\right)\alpha \left(\sigma \left(t_{k}\right)\delta_{k}\right)c\left(t_{k}\right),$$(7)$$\hat{A}\left(r\right)=\sum_{k=1}^{K}\hat{T}\left(t_{k}\right)\alpha \left(\sigma \left(t_{k}\right)\delta_{k}\right)A\left(t_{k}\right),$$
where $\hat{T}\left(t_{k}\right)=\exp\left(-\sum_{k^{\prime }=1}^{k-1}\sigma (t_{k})\delta_{k}\right)$, α(x)=1−exp(−x), and $\delta_{k}=t_{k+1}-t_{k}$ is the distance between two adjacent sample points.

We train the network in a coarse-to-fine way, with photometric loss $L_{p}$ and affordance loss $L_{a}$:(8)$$L_{p}=\sum_{r\in R}\left[{∥{\hat{C}}_{f}\left(r\right)-C\left(r\right)∥}_{2}^{2}+{∥{\hat{C}}_{f}\left(r\right)-C\left(r\right)∥}_{2}^{2}\right],$$(9)$$L_{a}=-\sum_{r\in R}\left[\sum_{l=1}^{n}p^{l}\left(r\right)\log{\hat{p}}_{c}^{l}+\sum_{l=1}^{n}p^{l}\left(r\right)\log{\hat{p}}_{f}^{l}\right],$$
where *R* is the rays in the training batch. C(r) and $p^{l}\left(r\right)$ are the ground truth for RGB color and affordance probability at class *l* of the sampled ray *r*. ${\hat{C}}_{c}\left(r\right)$ and ${\hat{p}}_{c}^{l}$ are the predicted color and affordance probability at class *l* of the coarse network, while ${\hat{C}}_{f}\left(r\right)$ and ${\hat{p}}_{f}^{l}$ are the output of the fine network. $L_{p}$ and $L_{a}$ are chosen separately as mean square error loss and multi-class cross-entropy loss.

The final loss *L* is a weighted sum of the above two losses:(10)$$L=L_{p}+\lambda L_{a}.$$

During the training process, a single image is utilized to extract shape and appearance features in each epoch. Moreover, the single image used for encoding is selected randomly in the train set, and the image corresponding to the sample ray *r* is not chosen. This makes the shape and appearance encoding less relevant to the view direction and passively enhances the feature extraction capability of the encoder. This image sampling strategy enables our network to function with limited views.

### 3.2. Dataset Construction

Existing affordance datasets provide affordance annotations, but the camera poses are missing, and there is a lack of images with multiple perspectives on a specified object [3]. For the above reasons, we have created a new dataset specifically for assessing our proposed AFF-NeRF from two sources. First, we obtain diverse models from open-source resources and then import the models into CoppeliaSim [26]. Color and depth images with poses are randomly captured on the hemisphere by controlling the camera in CoppeliaSim, as shown in Figure 3. After that, the images are annotated with affordance. For other objects, we use a similar method to obtain the same configured dataset from the PartNet [27]. The dataset includes a total of six types of objects, among which are screwdrivers, eyeglasses, buckets, bottles, cabinets, and teapots, and 120 images were captured from random viewpoints of each object. Figure 4 shows some of the images in the dataset.

## 4. Experimental Evaluation

The main focus of this work is an affordance generation pipeline, working with a single input view. We take various category objects with different shapes to evaluate the proposed method. At the same time, we integrated the acquired 3D affordance model into the grasp generation algorithm to investigate whether affordance understanding improves grasp performance.

### 4.1. Experimental Setup

**Implementation Details** For the shape and appearance encoder SE and AE, we use the ResNet18 [25] backbone without pre-training. Both the shape and appearance code are 32-dimensional vectors. Following the original NeRF [5], the lengths of position encoding for 3D position and view direction are 10 and 4. For depth information in the dataset, we set the near and far sample boundaries to 0.1 m and 1.5 m. All the images were resized to 400×300 for all experiments. The model was implemented in PyTorch 1.12.1 [28] and trained on a single RTX3090 produced by Nvidia with 24GB memory. The system has an i9-10900K processor produced by Intel and 64 GB of RAM. The λ in the total loss was set to $4\times 10^{-4}$. The batch size of rays was set to 1024, while the learning rate for the whole network was set to $5\times 10^{-4}$.

**Comparison Methods** We compare our method with GSNeRF [29], Semantic-Ray [30], and modified PixelNeRF [31] on seen categories. The modified PixelNeRF renders new views with affordance using inputs from three views, while GSNeRF and Semantic-Ray are consistent with the original implementation. For unseen categories, we compare with Semantic-NeRF [6], GSNeRF, and Semantic-Ray. By limiting the objects in the training set, Semantic-Ray and GSNeRF can also work on the unseen categories.

Meanwhile, we choose to [32] evaluate the affordance understanding on grasp generation.

**Evaluation Metrics** Our approach focuses on affordance generation, and affordance refers to semantics in this paper. Therefore, we use the metrics for semantic segmentation to evaluate affordance. We use mean intersection over union (mIoU), mean pixel accuracy (MPA), and pixel accuracy (PA) to evaluate the quality of the rendered images with affordance. The detailed calculation of all metrics can be found in [4].

For grasp generation, we use the ϵ-metric [33] to evaluate, which is widely used to measure the grasp quality.

### 4.2. Experimental Results

#### 4.2.1. Results of Affordance Understanding

The first experiment confirms that our approach consistently produces accurate affordance for various category objects that are not in training sets with only a single view, without requiring retraining, fine-tuning, or any labels.

Table 2 shows the quantitative results of affordance generation on various category objects. Our approach achieves the highest average mIoU, pixel accuracy, and mean pixel accuracy compared to the baseline methods on the categories of scissors, pens, glasses, and screwdrivers. Our approach with single view input averages 20.6% improvement in mIoU, 1.0% improvement in pixel accuracy, and 25.1% improvement in mean pixel accuracy. The qualitative results are shown in Figure 5 and Figure 6. The results indicate a significant enhancement in mean pixel accuracy and mIoU. These results suggest that the proposed method exhibits a distinct advantage in multi-category affordance understanding. Our method is able to respond to small structures of the cross-category objects, such as pen caps, lenses, etc. In contrast, the baseline methods demonstrate affordance confusion when addressing small structures. We also conduct experiments on the object component in different positions. Figure 7 shows the generated affordance model. The result shows that our method can respond to the object in various states.

#### 4.2.2. Grasp Application

In addition to rendering new images, our method can also generate 3D models with affordance from a single image without extra manual processing. Affordance is strongly correlated with robot manipulation tasks, wherein the generation of various strategies for different structures is often required. To verify the quality of the 3D model with affordance generated by our method, we input it into the grasp proposal algorithm [32], scoring the grasps using the metrics introduced in [33]. To compare, we also used the same grasp proposal method on the original object model without affordance understanding and compared the generated grasps. Table 3 shows the quantitative results of grasp scores on screwdrivers and eyeglasses. The qualitative results are shown in Figure 8. Our approach enables the grasp proposal algorithm to generate higher-quality grasps.

## 5. Conclusions

This paper presented a novel NeRF method, AFF-NeRF, to address the problem of 3D affordance understanding for homogeneous objects. Our AFF-NeRF model decouples the shape and appearance of the object and extracts the corresponding code from only a single image. This allows our approach to adapt quickly to the various category objects in manipulation tasks. These codes are then integrated into the extended NeRF to render novel view images with affordance. To evaluate our approach, we also built a dataset containing different objects of various shapes with affordance annotation. We conducted experiments to validate the effectiveness of our AFF-NeRF model. The first experiment verified that our method achieves affordance rendering with novel objects without retraining or fine-tuning with single-view input. The experimental results demonstrate that our approach achieves significant improvements over the baseline and exhibits strong generalization to unseen objects.Meanwhile, our method has limitations when dealing with complex structures, partial occlusions, or extreme lighting. We will attempt to address these limitations in future work. Furthermore, integrating our method with a grasp generation algorithm reveals that the affordance model enhances the generation of high-quality grasps.

## Figures and Tables

**Figure 1 biomimetics-10-00410-f001:**
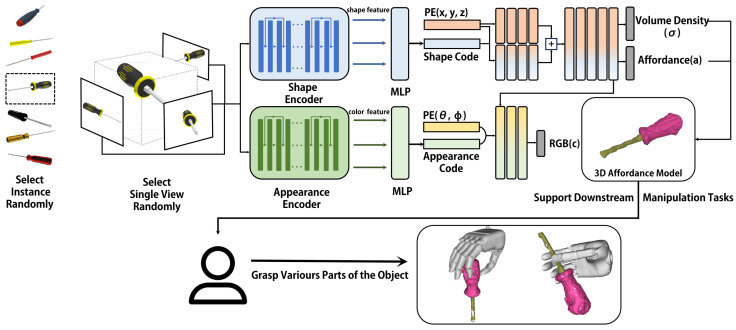
The pipeline of the proposed method. During training, our method first randomly selects a single image of the objects in the training set. Then, the images are fed into shape and appearance feature encoders. The 3D position (x,y,z) and viewing direction (θ,σ) are involved in the AFF-NeRF after position encoding (PE). The shape and appearance code are combined with position and viewing direction separately. Eventually, AFF-NeRF outputs volume density σ, affordance *a*, and color *c*. The 3D affordance models obtained from our method can be used in grasp generation algorithms to support downstream manipulation tasks.

**Figure 2 biomimetics-10-00410-f002:**
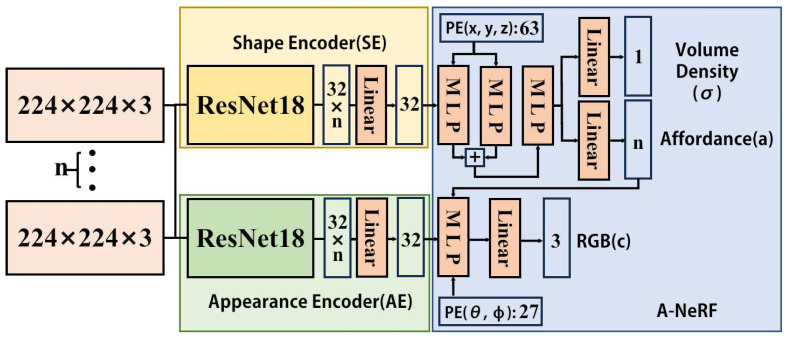
Network architecture. This figure shows a more detailed structure of the shape encoder and appearance encoder.

**Figure 3 biomimetics-10-00410-f003:**
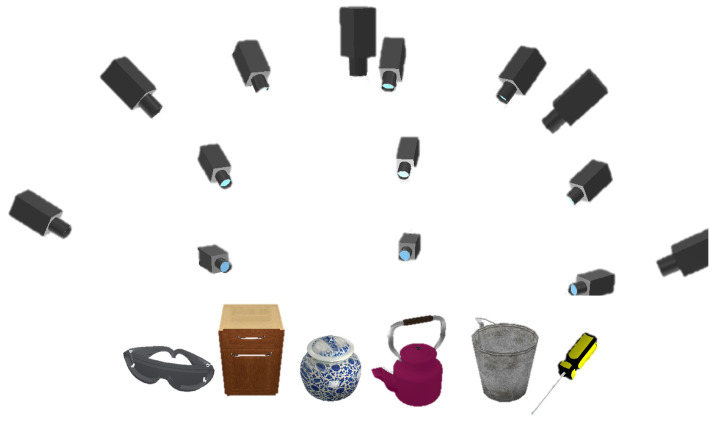
Color and depth images with poses are randomly captured on the hemisphere by controlling the camera in the simulation environment.

**Figure 4 biomimetics-10-00410-f004:**
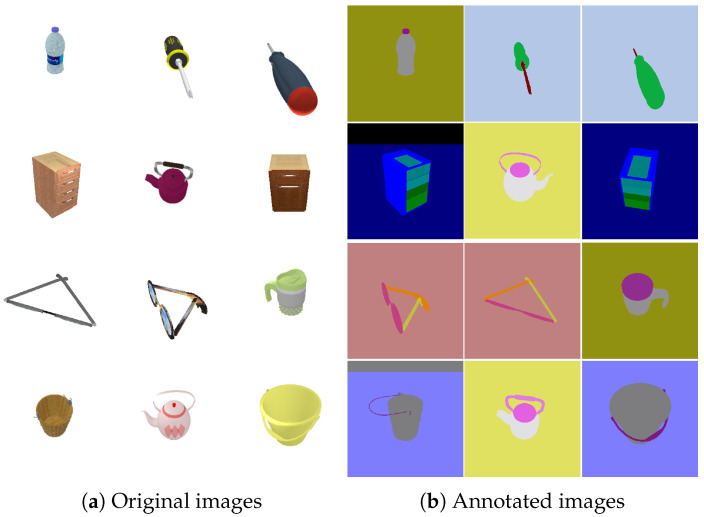
We have established a dataset containing objects of different categories with affordance annotation.

**Figure 5 biomimetics-10-00410-f005:**
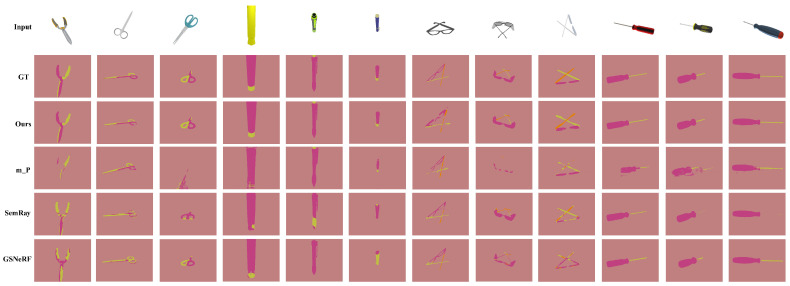
Results on the seen-category objects with single-view input. Our method is capable of generating affordance from novel perspectives of various categories of objects without retraining or fine-tuning. GT denotes the ground truth, m_P denotes the modified PixelNeRF, and SemRay denotes Semantic-Ray. With single image input, our method generates accurate affordance of novel perspectives, whereas the baseline yields affordance confusion.

**Figure 6 biomimetics-10-00410-f006:**
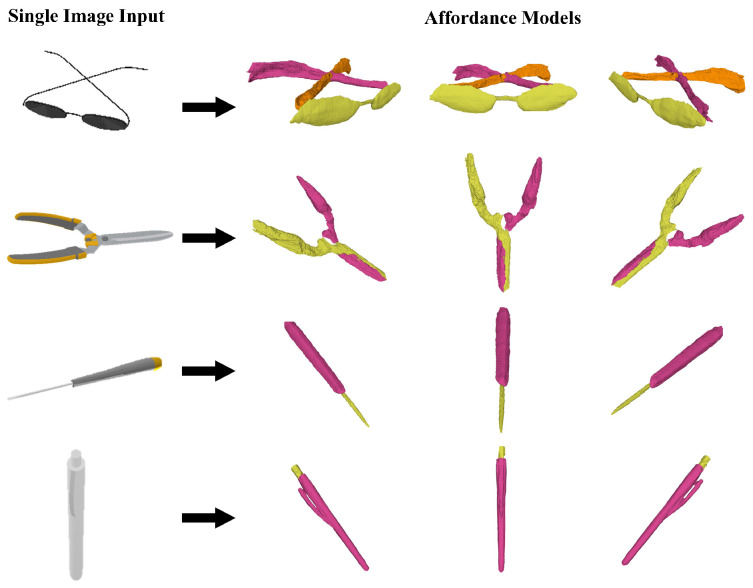
Affordance models generated by AFF-NeRF. Using single image inputs, AFF-NeRF affordance models of various seen-category objects are generated.

**Figure 7 biomimetics-10-00410-f007:**
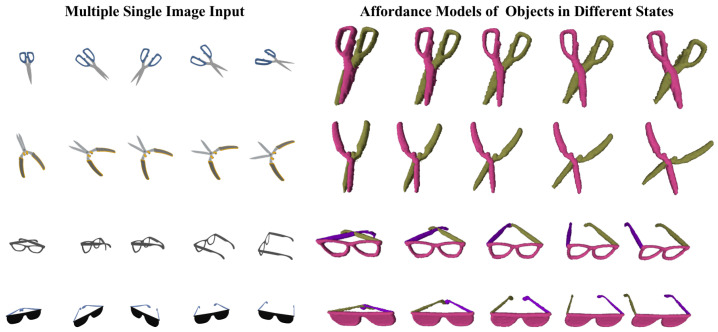
For articulated objects in the dataset, AFF-NeRF can generate affordance models of objects in various states.

**Figure 8 biomimetics-10-00410-f008:**
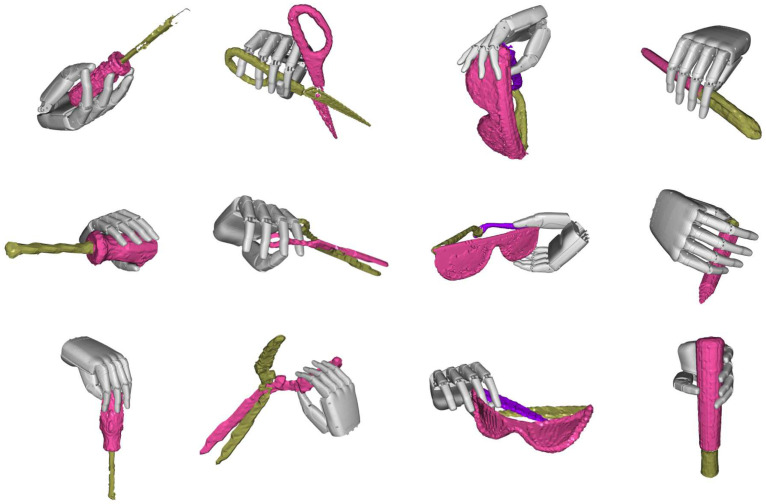
Visualization results of grasp generation algorithms with affordance model inputs.

**Table 1 biomimetics-10-00410-t001:** A comparison with prior works. Our approach can accomplish affordance understanding in a single view of the new object without additional training, unlike other methods.

	GIRAFFE [9]	EditNeRF [16]	CodeNeRF [15]	AFF-NeRF
Learns scene prior	✓	✓	✓	✓
Affordance outputs				✓
Allows zero-shot	✓		✓	✓
Without retraining	✓			✓

**Table 2 biomimetics-10-00410-t002:** Performance on seen categories of affordance understanding.

		Scissors	Pens	Glasses	Drivers	Average
mIoU	PixelNeRF_M	0.723	0.688	0.518	0.570	0.625
	SemRay	0.537	0.627	0.563	0.659	0.597
	GSNeRF	0.670	0.576	0.584	0.678	0.627
	Ours	**0.784**	**0.892**	**0.586**	**0.709**	**0.743**
PA	PixelNeRF_M	0.965	0.971	0.962	0.932	0.958
	SemRay	0.979	0.986	0.965	0.978	0.977
	GSNeRF	0.982	0.989	0.962	**0.982**	0.979
	Ours	**0.989**	**0.992**	**0.967**	0.977	**0.981**
MPA	PixelNeRF_M	0.729	0.403	0.547	0.613	0.573
	SemRay	0.645	0.656	0.682	0.690	0.668
	GSNeRF	0.770	0.759	0.687	0.684	0.725
	Ours	**0.940**	**0.799**	**0.733**	**0.777**	**0.812**

**Bold** denotes the best. PixelNeRF_M denotes the modified PixelNeRF. SemRay denotes Semantic-Ray.

**Table 3 biomimetics-10-00410-t003:** Grasp results with affordance model and original model.

	Screwdriver	Pen	Scissor	Eyeglass
Original Model (no affordance)	0.241	0.197	0.242	0.202
Affordance Model	0.272	0.198	0.306	0.242

## Data Availability

The data supporting this study’s findings are available from the corresponding author or the first author upon reasonable request.

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
