# Peer review of "Bio-Inspired 3D Affordance Understanding from Single Image with Neural Radiance Field for Enhanced Embodied Intelligence"

_biomimetics, 2025, doi:10.3390/biomimetics10060410_

Round 1
Reviewer 1 Report
Comments and Suggestions for Authors
This manuscript can be accepted after minor revision. There are no text in the introduction that is related with " Bio-inspired". The number of the images used for training should be mentioned in the methodology section. The computer/main hardwares used in the experimental setup should be mentioned. This paper " Shih-Yang Su, Frank Yu, Michael Zollhöfer, and Helge Rhodin. "A-NeRF: Articulated Neural Radiance Fields for Learning Human Shape, Appearance, and Pose", NeurIPS, 2021" also used the name of " A-NeRF". Is there anything related? The advantages and drawbacks of this method can be discussed in the end of the discussed section. The English should be improved a lot. I cited several examples as follows: (1) Our method extracts shape and appearance features of various
objects using deep residual networks... (2) "The results supported that our approach has significant improvements over the baseline and generalizes well to unseen objects. Moreover, we combine the method with grasp generation algorithm. The results show that the affordance model is beneficial in generating high-quality grasps "
Reviewer 2 Report
Comments and Suggestions for Authors
The paper presents the A-NeRF method for generating 3D affordance models from a single image. The authors use neural emission fields (NeRF) extended with modules for extracting shape and appearance features of objects. The paper has an interesting idea of combining NeRF with the affordance task, but I have some questions for the authors:
1) The paper only studies 6 types of objects. It is unclear how the method scales to more diverse or complex categories. For example, how does the method cope with objects that have a complex or ambiguous structure, such as multifunctional tools? There is also a question about the representativeness of the dataset, it would be interesting to see other variants of models in open positions (for example, scissors).
2) How does the method work with partially occluded objects or objects in difficult lighting conditions?
Overall, the paper is very interesting, and using NeRF to generate affordances is a good idea, especially in the context of robotic manipulation. The graphs are ok, the methods are described well, the results are clearly presented.
I believe that the article is ready for publication after minor revisions.
